# Olive Tree in Circular Economy as a Source of Secondary Metabolites Active for Human and Animal Health Beyond Oxidative Stress and Inflammation

**DOI:** 10.3390/molecules26041072

**Published:** 2021-02-18

**Authors:** Rosanna Mallamaci, Roberta Budriesi, Maria Lisa Clodoveo, Giulia Biotti, Matteo Micucci, Andrea Ragusa, Francesca Curci, Marilena Muraglia, Filomena Corbo, Carlo Franchini

**Affiliations:** 1Department of Bioscience, Biotechnology and Biopharmaceutics, University Aldo Moro Bari, 70125 Bari, Italy; rosanna.mallamaci@uniba.it; 2Department of Pharmacy and Biotechnology, Food Chemistry & Nutraceutical Lab, Alma Mater Studiorum-University of Bologna, 40126 Bologna, Italy; roberta.budriesi@unibo.it (R.B.); giulia.biotti@studio.unibo.it (G.B.); matteo.micucci2@unibo.it (M.M.); 3Interdisciplinary Department of Medicine, University Aldo Moro Bari, 702125 Bari, Italy; marialisa.clodoveo@uniba.it; 4Department of Biological and Environmental Sciences and Technologies, Campus Ecotekne, University of Salento, 73100 Lecce, Italy; andrea.ragusa@unisalento.it; 5Department of Pharmacy-Drug Sciences, University Aldo Moro Bari, 70125 Bari, Italy; francesca.curci@uniba.it (F.C.); marilena.muraglia@uniba.it (M.M.); carlo.franchini@uniba.it (C.F.)

**Keywords:** *Olea europea* L., olive oil, olive mill wastewater (OMW), olive leaf extract (OLE), hydroxytyrosol, oleuropein, polyphenols, pit, by-products

## Abstract

Extra-virgin olive oil (EVOO) contains many bioactive compounds with multiple biological activities that make it one of the most important functional foods. Both the constituents of the lipid fraction and that of the unsaponifiable fraction show a clear action in reducing oxidative stress by acting on various body components, at concentrations established by the European Food Safety Authority’s claims. In addition to the main product obtained by the mechanical pressing of the fruit, i.e., the EVOO, the residual by-products of the process also contain significant amounts of antioxidant molecules, thus potentially making the *Olea europea* L. an excellent example of the circular economy. In fact, the olive mill wastewaters, the leaves, the pomace, and the pits discharged from the EVOO production process are partially recycled in the nutraceutical and cosmeceutical fields also because of their antioxidant effect. This work presents an overview of the biological activities of these by-products, as shown by in vitro and in vivo assays, and also from clinical trials, as well as their main formulations currently available on the market.

## 1. Introduction

Olive oil is the main product obtained from olives, fruits that come from the *Olea europaea* L. evergreen trees [1]. Olive oil is a characteristic element of the Mediterranean Diet (MD) because of the health-beneficial effects deriving from its chemical composition [2,3,4] as well as its appreciable taste and usefulness in flavoring a large variety of foods. In particular, the constituents of both lipidic and unsaponifiable fractions in extra-virgin olive oils (EVOOs) have been demonstrated to be able to reduce oxidative stress by acting on various biomolecules in the body, as also stated by the European Food Safety Authority (EFSA) [5].

During the production process of EVOOs, olive milling yields a mixture of olive paste and water. Subsequent malaxation of the olive paste allows for the separation of three phases: (i) the olive oil, (ii) a solid residue, and (iii) the olive mill wastewater (OMW). The last two components are produced in large quantities, and they are considered an agro-industrial waste whose disposal represents an important environmental problem as the plant material is usually subjected to microbial deterioration [6,7].

Currently, in the linear economy, agricultural by-products are mainly used as combustion feedstock for biofuels (Figure 1) [8,9].

The most important biomasses are residues from wood working (wood shavings and sawdust) or forestry activities, wastes from farms and agro-business, the organic fraction of municipal solid wastes, and plants deliberately grown for energetic purposes. Similarly, pruning wastes from olive trees are also used as biomass. However, in coherence with the “circular economy” principle, it is important to valorize these waste products containing high levels of secondary metabolites, thus accelerating the implementation of the “Transforming our world: the 2030 Agenda for Sustainable Development” [10,11].

Nevertheless, the transition from linear to the circular economy requires a cultural and structural change: a deep revision and innovation of production, distribution, and consumption models [12]. Furthermore, from a circular economy perspective, the added value of materials and energy must be maintained for as long as possible over multiple productions and use cycles, representing a new opportunity also for seasonal sectors, such as the EVOOs manufacturing industry.

Olive mill waste, olive pomace (exhausted pulp, kernel, and seeds), and vegetative water are significant by-products of the olive oil-producing countries in the Mediterranean basin, with a high environmental impact if not properly treated. In addition, these wastes are rich in high-value compounds, which can be either used directly after extraction or exploited as ingredients with different applications, e.g., as food supplements, nutraceuticals, cosmeceuticals, and animal feed.

The transition from linear to the circular economy, largely desired from stakeholders in the olive oil sector, requires a multidisciplinary approach that exploits know-how harmonically from different fields (Figure 2).

This transition would also bring an additional value, represented by the possibility that each oil mill can integrate new processes with the pre-existing ones, with the resulting economic advantages, guaranteeing both product diversification and fair income for all stakeholders, who are currently threatened by the increasing oil price trend and the emerging Xylella pandemic.

In this review, we focused our attention on the secondary metabolites contained in waste materials derived from the olive oil production process and their ability to reduce oxidative stress, both in vitro and in vivo. Particular attention has been paid to their potential exploitation in the circular economy by obtaining new high-value ingredients for health-related products (nutraceuticals, pharmaceuticals, and cosmeceuticals).

## 2. *Olea europea* L.: Overview on Its Chemical Compounds

The most represented chemical classes in *Olea europea* L. tree are mainly classified as nonpolar compounds (present in the lipophilic oil fraction, such as squalene, tocopherols, sterols, and triterpenic compounds) and polar phenolic compounds [13].

Among the polyphenolic compounds, the most abundant and studied in olives are tyrosol (TY), hydroxytyrosol (HT), oleuropein (OL), oleocanthal, and verbascoside (Figure 3).

The secondary metabolites from *Olea europea* L. have high biological value, and they are present in different concentrations in the various parts of the olive plant (Table 1); as such, many of them are present in the derived EVOOs, but they can also be found in the waste products from the production process.

For this reason, all the materials involved in olive oil manufacturing represent a precious reservoir that could supply extracts reusable for health purposes. The most studied secondary metabolites are the polyphenols (or biophenols, as they are often referred to in EVOOs) that represent a group of molecules with one or more phenolic rings [14]. These compounds can be defined as nutraceuticals for their biological/pharmacological actions [15], mostly derived from their antioxidant properties, that play a protective role against oxidative stress [16] and extend the shelf-life of olive oil [17].

The antioxidant activity is mainly due to five classes of polyphenols identified as simple phenols, phenolic acids, secoiridoids, flavonoids, and lignans [18]. Among these, OL represents the principal biophenol in the olive leaf [19], followed by other constituents such as verbascoside, luteolin-7-*O*-glucoside, apigenin-7-*O*-glucoside, and TY [20]. Their antioxidant activity is even higher than that of antioxidants, such as vitamins E and C [21].

## 3. *Olea europea* L. By-Products for Human Health

The plant *Olea europea* L. is a genus that comprises more than 40 species. To this genus belong plants that are typical of temperate regions in the European continent, Asia, and Africa. We focused on *Olea europaea* L. because it is the only species used for obtaining oil by pressing their fruit (i.e., the olive). On the other hand, other species, such as *O. capensis*, *O. dioica*, *O. brachiata*, and *O. obvata*, are not used for oil production. Currently, there are almost 400 cultivars of *Olea europaea* L. used all over the world, of which about 100 are planted in Italy.

Olive products, such as olive oil and table olives, are functional foods because of their beneficial effects, mostly due to mono- and poly-unsaturated fatty acids and, last but not least, the presence of polyphenols and other secondary metabolites. During the olive oil production process, some polyphenols remain in the oil-water emulsion, but most of them, being hydrophilic, end up in the OMW. Sometimes olive leaves are also added to the olives before milling in order to enrich the resulting oil in polyphenols. In addition, many production factors (e.g., cultivar, ripening time, and extraction method), as well as environmental factors (e.g., climate, precipitations, and age of the trees), are responsible for the different content and composition of polyphenols in oil [22].

The nutritional and health-promoting effects of olives and olive oils are well-established and recognized [23], such as their antioxidant [24,25], anti-inflammatory [26,27], cardioprotective [28,29], anticancer [30], antidiabetic, and neuroprotective effects [31,32]. Thanks to these properties, these compounds positively contribute to the beneficial effects of the MD [33]. To confirm the significant role of olive oil components as responsible for the benefits of the MD, Fernandes et al. examined the outcome from randomized controlled trials on the effect of regular dietary EVOO intake on inflammatory markers [34]. Recently, Storniolo and co-workers demonstrated that the role of oleic acid in the colon cancer cells growth is reverted in the presence of olive oil representative minor components, suggesting that the consumption of seed oils, high oleic acid seed oils, or olive oil will probably have different effects on colorectal cancer [35,36]. The presence of secondary metabolites also in the by-products of olive oil production makes OMW, leaves, pomace, and kernels raw materials exploitable in the nutraceutical, food, cosmetic, feed, and energy sectors. The scientific evidence related to the health-promoting effects of these by-products is detailed below.

### 3.1. Secondary Metabolites in Olive Mill Wastewater

OMW is a by-product of olive oil production, rich in water-soluble bioactive compounds that could be separated by industrial membrane technology [37]. This procedure, based on the different capabilities of the substances in a mixture to cross the polymeric or inorganic semipermeable membrane at different rates, allows a cost-effective purification of the OMW phenolic pool because of the low operative temperature needed [38].

Nanofiltration has also been successfully employed for concentrating phenolic compounds extracted from the same raw material. The extracts are fractionated across different membranes to get microfiltration, followed by ultrafiltration and nanofiltration [39]. The total phenolic content is then analyzed using high-performance liquid chromatography (HPLC) [40].

OMW has long been considered a waste whose disposal requires high economic costs. Recently, numerous studies have shown its content in polyphenols and other biologically important molecules, shifting its perspective from waste to an economical and natural source of antioxidants [41]. The typical composition of OMW is reported in Table 2. As can be observed, OL, abundant in leaves, is absent in OMW, while are present several low molecular weight phenolic compounds, such as TY and HT, which are formed by enzymatic hydrolysis during the milling process. Phenolic compounds with molecular weights in the range of 600–5000 Da and other molecules, such as verbascoside, its isomers, and oxidation products, as well as higher molecular weight phenols deriving from the oxidative polymerization of hydroxytyrosol and elenolic acid, are also present [42,43].

OMW also contains significant amounts of monosaccharides, such as glucose, galactose, arabinose, rhamnose, and galacturonic acid, and polysaccharides, whose prebiotic and antioxidant activities have been evaluated [44,45,46,47]. Among simple sugars, arabinose, in particular, showed to be able to reduce the concentration of hydroxyl radicals by chelating Fe^2+^ ions [48] significantly. Many studies on various matrices also showed the antioxidant capacity of polysaccharides [49,50,51]. Therefore, the remarkable antioxidant activity of OMW can be attributed not only to its polyphenolic content but probably also to the polysaccharide and protein content. Furthermore, polysaccharides in OMW assimilated to dietary fibers owed additional biological and physiological functions, such as antimetastatic, immunostimulating, and anti-ulcer activity, as well as reduction of serum cholesterol, inhibition of hyaluronidase, and release of histamine [45,52].

### 3.2. Biological Activity of Olive Mill Wastewater Extracts

Recently, many research groups have tested OMW, in which both HT and its precursors are much more concentrated with respect to olive oil, on numerous biological targets. A study on two OMW mixtures with a polyphenol content of 100 and 36 g/kg (MOMAST^®^ HY100 and MOMAST^®^ HP30, respectively) found a significant antioxidant and anti-inflammatory effect in an ex vivo model of rat colon, liver, heart, and prefrontal cortex [53]. After treatment, the levels of the several inflammatory markers, i.e., prostaglandin (PGE_2_), lactate dehydrogenase (LDH), nitric oxide synthase (iNOS), COX-2, and TNFα, decreased drastically.

Other studies on purified extracts of OMW have shown additional anti-angiogenic and chemopreventive effects, both in vitro and in vivo [54,55], as well as inhibition of the proliferation, migration, and invasion of endothelial cells [56]. Furthermore, the antiproliferative activity of OMW against MDA-MB-231 breast cancer cells has been also demonstrated [57]. Chemopreventive effects of OMW rich in HT have also been observed in HL60 human promyelocytic leukemia cells, HT-29, and DLD1 colon adenocarcinoma cells, reducing cell proliferation by inducing apoptosis [58].

Noteworthy, OMW extracts were shown to have neuroprotective effects both in vitro and in vivo on dissociated brain cells (DBC) of NMRI mice. Even though the mechanism of action is not yet fully understood, it is likely that the biological effect is due to its antioxidant and anti-inflammatory action by inhibiting lipid peroxidation and restoring glutathione concentrations. The secoiridoids in OMW are also responsible for the beneficial effects in delaying cellular aging in neurodegenerative disease. For example, they were able to interfere with aggregation of amylin [31], tau [59], and Aβ peptides in vitro [60], in *C. elegans* [61], and in the mouse model TgCRND8 of Aβ deposition [62], which appears to be dose-dependent [63]. Neuroprotection exerted by biophenols has also been demonstrated in neuroblastoma cells by reducing the oxidative stress induced by H_2_O_2_ and the toxicity induced by copper (Cu) [64]. The cytoprotective effects of formulations containing both HT and OMW were compared on the same cell line by inducing toxicity after 24 h of exposure to cadmium (Cd), mercury (Hg), and lead (Pb), showing that the polyphenols could slow down or even halt the progression of the disease aggravated by heavy metals [65].

OMW phenolic compounds were also able to reduce risk factors for coronary heart disease and stroke prevention [66]. Furthermore, Storniolo et al. highlighted that HT and other polyphenols play an important role in preventing the negative consequences of diets rich in fats and/or sugars [67]. They showed that treatment with HT or OMW could reduce significantly the level of nitric oxide (NO) and the increase of endothelin-1 (ET-1) by modulating the intracellular levels of Ca^2+^ and the endothelial phosphorylation of nitric oxide synthase, changes induced by high levels of glucose and free fatty acids (as in diabetic patients).

Due to its antioxidant properties, OMW could easily find applications in the food, pharmaceutical, and cosmetic industries. For example, it could be used to better preserve the quality and shelf life of food [68,69]. Production of functional foods from OMW extracts represents a crucial alternative to transform this agro-industrial waste into a useful and relevant ingredient [70,71]. An interesting approach for fortifying food products with phenolic substances involves their direct addition [72]. In this regard, OMW phenolic extracts were added to milk to study their effect in modulating the Maillard reaction when milk is heated at very high temperatures. The authors reported that the phenolic extracts were able to trap the reactive carbonyl species responsible for the unpleasant taste and to inhibit the formation of Amadori products [73]. The use of OMW phenolic compounds in milk-based beverages has also been reported to improve their nutritional properties; in fact, as the concentration of more complex phenolic compounds decreased during storage, the level of HT increased due to the hydrolysis of its precursors [74].

In light of all these pieces of evidence, it can be hypothesized that the phenolic compounds present in OMW, such as HT and OL, could soon be considered raw materials for nutraceutical supplements or formulations. Several HT-containing products, such as Mediteanox^®^, Hydrox^®^, and Hytolive^®^, are already on the market in pharmaceutical forms, such as capsules, elixirs, creams, and even in EVOOs with a very high HT content (over 500 mg/kg). Hydrox^®^ and Hytolive^®^ have been licensed as “generally recognized as safe (GRAS)” ingredients. Pure synthetic hydroxytyrosol, marketed by SEPROX BIOTECH, has also achieved this status and has recently been proposed in the EU for Novel Food. Some of these products have already been tested successfully [75,76].

Polyphenols are massively used as cosmetic ingredients. In fact, it is known that UV irradiation and oxidative stress are the main causes of extrinsic aging and of diseases such as skin cancer [77]. The protective action against UV damage, inhibition of the antimicrobial activity of dermal proteinases, and the anti-carcinogenic action have been demonstrated in vitro on skin cell lines. These findings could be exploited for preparing novel topical formulations. The protective effect exerted by polyphenols against lipid oxidation on cell membranes, an effect that mimics the protection from the oxidation of oil lipids by polyphenols, can also prevent oxidative phenomena in the formulation during storage [78,79]. The topical application of active antioxidant ingredients can support the skin’s own antioxidant system against oxidative stress and may protect the skin from long-term photoaging.

High reactive oxygen species (ROS) production also results in the expression of collagenase (MMP-1) and elastase, leading to accelerated degradation of the corresponding proteins. Lee et al. showed that polyphenols effectively inhibit elastase and hyaluronidase, exerting an anti-aging effect [80]. Treatment of HaCaT keratinocytes with polyphenolic extracts resulted in a reduced formation of intracellular ROS after UV irradiation [81,82,83]. Finally, Potapovich et al. showed that post-treatment with polyphenols of normal human epidermal keratinocytes (NHEKs) after UV exposure was effective in abolishing the overproduction of peroxides and inflammatory mediators [84]. In this regard, considering the composition of OMW, it would be interesting to investigate more in detail this aspect as OMW could also present similar anti-aging effects.

### 3.3. Secondary Metabolites and Biological Activity of Olive Pomace Extracts

The main destiny of olive pomace in the linear economy is its transfer to an olive pomace factory, where it is dried and then used for extracting with organic solvents (usually hexane) the residual fat (crude pomace oil), which will be then rectified before marketing. In recent times, the price of pomace oil has significantly dropped, making, in some cases, its extraction uneconomical. In addition, the sector had already been experiencing great difficulties because of the increased water content in virgin pomace due to the increasingly widespread use of two-phase decanters. The opportunity to consider the pomace not only as a source of fats but also of a complex mixture of bio-compounds can be advantageous for both the environment and the miller’s income. In fact, the possibility to implement a new production process into the mill could potentially provide an additional source of profit for both olive oil producers and olive millers, thus closing the supply chain at the production site.

In this regard, Nunes et al. investigated the chemical composition of the bioactive compounds in olive pomace (e.g., fatty acids, vitamin E, and phenolic compounds) and its nutritional profile and they also developed a sustainable process for extracting the antioxidants (the Multi-frequency Multimode Modulated Ultrasonic technique) [85]. Moreover, they discovered that the vitamin E profile of the olive pomace contained high amounts of α-tocopherol (2.63 mg/100 g), although β- and γ-tocopherol and α-tocotrienol were present in lower concentrations (less than 0.1 mg/100 g of pomace). Oleic acid was the most abundant lipid, followed by palmitic, linoleic, and stearic acid (10%, 9%, and 3%, respectively), while the polyphenols were mainly represented by HT and comsegoloside (making together about 79% of the total content). A year before, Goldsmith et al. had already tested an innovative ultrasound method with the aim to increase the aqueous extraction of phenolic compounds from olive pomace [86]. Application of a Design of Experiments allowed the authors to find the optimal extraction conditions, although the process was not very efficient (2 g of dried pomace/100 mL of water at 250 W for 75 min at 30 °C), thus limiting the technological transfer to an industrial level.

Recently, the interest in the water-soluble fraction from olive pomace is high because numerous authors are demonstrating the potential beneficial effects of the contained sugars, polyphenols, and minerals. Ribeiro et al. investigated the effect of the gastrointestinal tract on its bioactive composition, demonstrating that about 50% of the water-soluble compounds remained active, especially of HT and potassium [87]. In addition, the recovered antioxidant activity in the serum was about almost 58%, and more than 50% of the initial α-glucosidase inhibition activity was maintained, as well as its ACE inhibitory activity. The colon-available fraction presented a substantial concentration of polyphenols and minerals, evidencing that OMW liquid-enriched powder could be potentially useful to prevent both cardiovascular and gut diseases. The potential effects in terms of liquid-enriched powder marketing are interesting, although further studies are needed to confirm preliminary results.

Other authors investigated even simpler techniques to make the extraction process more convenient. Cea Pavez et al. exploited pressurized liquid extraction (PLE) for extracting phenolic compounds from olive pomace [88]. Despite the extraction protocol showed great compositional variability of the obtained mixtures depending on the experimental conditions used; after optimization, PLE allowed the obtaining of a higher polyphenolic content compared to the traditional extraction method (1659 mg/kg and 282 mg/kg, respectively), also yielding three- and four-times higher concentrations of secoiridoids and flavonoids, as well as a significant HT enrichment.

In the context of environment-friendly green technologies, the use of deep eutectic solvents (DESs) has been gaining prominence in recent years. DESs have several advantages, including very low toxicity, ease of preparation, low cost, high biodegradability, and stability in the presence of water. In 2018, Chanioti et al. employed natural deep eutectic solvents (NADES) constituted by choline chloride with citric acid, lactic acid, maltose and glycerol, and water combined with homogenization (HAE), microwave (MAE), ultrasound (UAE), or high hydrostatic pressure (HHPAE) [89]. Choline chloride with citric acid and lactic acid showed the best extraction efficiency in terms of total phenolic content and antioxidant activity of the extracts, while HAE proved to be the best extraction technique. Extracts with NADES were generally richer in polyphenols compared to conventional solvent extraction procedures, and HPLC analysis confirmed that proposed methods are effective and sustainable alternatives for their extraction from natural sources.

The exploitation of compounds with high biological and commercial value is certainly the direction in which to push the transition of the oil sector. Many studies agree on the beneficial properties of these substances. Vergani et al. carried out a study on the biological effects of polyphenols extracted from olive pomace and on the effects of single phenolic compounds present in the extract (i.e., TY, apigenin, and OL) in protecting hepatocytes against fat excess and oxidative stress [90]. The polyphenols were extracted in ethanol/water (50:50 *v*/*v*) at high pressure-temperature (25 bar, 180 °C for 90 min), obtaining a total concentration of 5.77 mg of caffeic acid equivalent/mL. In order to test the biological effects of the extract, FaO cells exposed for 3 h to a mixture of oleate/palmitate (2:1 molar ratio) were used as a model for hepatic steatosis. The cells were incubated with TY, apigenin, or OL (10, 13, and 50 μg/mL, respectively), and the content of intra- and extra-cellular triglycerides (TGs) and other oxidative stress markers measured after 24 h. The preliminary results showed that olive pomace extract ameliorated lipid accumulation and lipid-dependent oxidative unbalance, suggesting them as potential therapeutic agents. The direct correlation between an MD supplemented with EVOO and a reduced prevalence of hepatic steatosis in older individuals at high cardiovascular risk was recently investigated in a clinical trial comprising one hundred men and women (mean age: 64 ± 6 years old) at high cardiovascular risk (62% with type 2 diabetes) [91].

The biological activity of polyphenols recovered from olive oil by-products was also investigated by Romani et al., who studied the cardioprotective effects of hydroxytyrosol, oleuropein, oleocanthal, and lignans in the MD [92]. Moreover, recent European projects, such as EPIC (European Prospective Investigation into Cancer and Nutrition) and EPICOR (long-term follow-up of antithrombotic management patterns in acute coronary syndrome patients), focused on the functional and health-promoting properties of EVOOs, showing the relationship between cancer and nutrition and the existent link between the consumption of EVOO, fresh fruits, and vegetables, and the incidence of coronary heart diseases. Results evidenced that both the EVOO and the by-products of the olive oil extraction process are precious sources of bioactive compounds that can be recovered applying green technologies and used for food, agronomic, nutraceutical, and biomedical applications, in agreement with the circular economy strategy.

### 3.4. Secondary Metabolites in Olive Leaves

Leaves represent an important quote of the total harvest weight. Therefore, it is important to develop efficient extraction methods that can assure high yields of polyphenols, secoiridoids, and other bioactive molecules that can be exploited in nutraceutical products, cosmetics, and functional foods (see Table 1). *Olea europaea* L. leaves are a potentially inexpensive, renewable, and abundant source of biophenols [93]. The importance of this agricultural and industrial waste needs to be emphasized and better understood, considering the benefits that we can get from it in health terms and also regarding the environment. Due to its antioxidant, antimicrobial, and anti-inflammatory effects, olive leaf extract (OLE) is considered a natural supplement. Several studies already showed the pharmaceutical and nutraceutical potentials of the secondary metabolites extracted from olive leaves. Microfiltration, ultrafiltration, and nanofiltration are all techniques able to provide OLEs with high amounts of polyphenols that could be exploited by cosmetic, food, and pharmaceutical industries.

Because natural active compounds are safer to use than synthetic chemicals, there is a growing interest in extracting oleuropein from olive leaves. However, the high operational cost, as well as the toxicity and flammability of the organic solvents usually employed, limits their exploitation. Nevertheless, the utilization of novel techniques, e.g., NADES, might bring a change [94]. In order to use these extracts for nutraceutical and pharmacological purposes, another crucial point to address is their bioavailability. In fact, when assumed orally, secondary metabolites in olive leaf should resist the gastric acid in the stomach before reaching the bloodstream. However, it has been observed that the amount of OL and verbascoside at the end of the digestion processes are almost negligible, mainly due to their chemical instability [95]. On the other side, luteolin-7-*O*-glucoside (Figure 4) was fairly resistant to digestion and, therefore, it can be considered an interesting polyphenol for oral administration.

The same study also analyzed if different extraction methods could influence the total amount of obtained polyphenols, either processing the olive leaves by freeze-drying at −20 °C or by hot air drying (70–120 °C), although the final concentration was nearly the same.

In order to determine the total polyphenol content of the leaves, a wide research study was performed on seventeen cultivars planted in Iran, including some varieties that are also present in Italy [96]. The total phenolic content and antioxidant activity of the leaves’ extracts were determined, showing that the Coratina cultivar has one of the highest content of polyphenols and the maximum radical scavenging activity. The OLE composition was mainly characterized by vanillin, rutin, luteolin 7-*O*-glucoside, oleuropein, and quercetin. High OL concentrations were also detected in other cultivars, such as the Mishen, Beleidi, Kalamon, and Roghani cultivars, while it was not detected in the Conservolea, Amigdalolia, Leccino, and Fishomi cultivars (Table 3).

Itrana, Apollo, and Maurino cultivars were the ones with the highest content of polyphenols, mainly quinic acid, oleuropein, and luteolin 7-*O*-glucoside, and also the ones with the strongest antioxidant activity. Italian olive cultivars, namely Dritta, Leccino, Caroleo, Coratina, Castiglionese, Nebbio, and Grossa di Cassano were also studied to determine their OL concentration in the extracts. Leaves from Nebbio, Grossa di Cassano, and Castiglionese olive trees revealed the highest oleuropein content. On the other hand, Caroleo, Leccino, and Dritta leaf extracts showed the lowest OL amounts.

Concluding, polyphenols have a wide range of bioactivities, and the olive leaf extracts could be either used as such in cosmetics, or they could be mixed with olives that are too ripe to produce oils with great resistance to oxidation, thus using them directly as olive oil supplements [97]. Alternatively, their phenolic extracts could be employed to produce dietetic tablets and food supplements, pharmaceuticals, and also to improve the shelf-life of foods.

In general, green leaves seem to have a higher OL content compared to the yellow ones [98]. Since leaves represent a significant part of the total harvest weight, it is of paramount importance to exploit them in the best way possible, as already stated, according to a circular economy approach.

### 3.5. Biological Activity of Olive Leaf Extracts

Olive leaves have been widely used in popular medicine to treat diseases like fever and other inflammation-related situations. The ancient Greeks and Romans used OLE as a natural remedy for treating hypertension. Leaves were also used in the past to prepare infusions.

It has been shown that olive leaf extract can lower blood pressure in animal models, alleviate arrhythmia, and exert spasmolytic activity on intestinal muscle [99]. Several studies attest to the antihypertensive effect of olive leaf extract by reducing systolic and diastolic blood pressure and even improving plasma TGs and LDL levels. Moreover, the antihypertensive effect did not show side effects on liver or renal functions in subjects with stage-1 hypertension, attesting its potential use as a preventive nutraceutical for chronic diseases [100].

Among the *Olea europea* L. polyphenols, oleuropein is a secoiridoid present as glucosylated derivatives in the olive fruit, while its dihydroxytyrosol and non-glucosylated secoiridoids were found in the leaf. OLE is a natural supplement that can be used either alone or in combination with other extracts, mainly in formulations that do not require a medical prescription. To support the key nutraceutical role of EVOO in the MD, Storniolo et al. recently analyzed whether it induced changes on endothelial physiology elements, such as NO, ET-1, and ET-1 receptors, which are involved in controlling blood pressure [101]. Some in vitro studies confirmed this action by analyzing a commercial extract on cardiomyocytes of rabbits’ hearts. As a result, OLE caused a concentration-depended decrease in systolic left ventricular pressure and heart rate, as well as an increase in relative coronary flow, maybe because of the direct and reversible suppression of the L-type calcium channel [102].

The vasorelaxant activity of OLE has also been investigated on aorta sections in addition to the inotropic and chronotropic effects measured on atria [103]. The vasorelaxant activity was related to the mechanism involving voltage and receptor operated Ca^2+^ calcium channels. The calcium antagonist activity is always to be considered in addition to the antioxidant activity and to other mechanisms involved in the same pharmacological direction, such as the direct effect on endothelium cells. The leaf extract was also shown to act by reducing the spontaneous contractility of the vessels, thus indirectly acting positively on the pressure exerted by the blood flow on the vessels [104].

OLE and *Hybiscus sabdariffa* L. flower extracts also showed calcium antagonistic properties [103]. Before the idea of formulating a nutraceutical product that synergizes the two activities, several in vitro and *ex-vivo* studies were conducted to verify the antagonist action directed to calcium channels. These two extracts have already been developed in a nutraceutical product, registered as “Pres Phytum” and already commercialized in Italy. In particular, the biological activity of the nutraceutical formulation led to vasorelaxant effects on smooth muscles in different districts of the body (IC_50_ 2.38 mg/mL) and to a negative chronotropic effect (IC_50_ 1.04 mg/mL) that could be exploited in the treatment of preclinical hypertension, without leading to a negative inotropic effect.

As we know, natural molecules do not only have an antioxidant effect; but we have to study and analyze their multitarget profile in order to understand what their real potential is [105]. Olive leaves phenols also reduce blood pressure with NO bioavailability modulation that is increased after a 28 day long dietary assumption [106]. Oleuropein and hydroxytyrosol induced the NO synthase and also had effects on NADPH, which also augmented the quote of superoxide.

OLE has also been studied in humans, and results are significantly positive in terms of cardioprotection [107]. When assumed as a dietary supplement (chronic consumption), the phenolic compounds contained in the olive leaf extract led to a reduction in LDL and TGs concentration that could be attributed to the antioxidant and calcium antagonist properties, but, in diabetic people, they also led to a reduction of the glucose concentration, probably because of the α-amylases inhibition, and to a reduction of glycosylated hemoglobin (HbAlc) and plasma insulin [106,108].

The anti-inflammatory effect on monocytes, the reduction of adhesion molecules, such as ICAM-1 and VCAM-1, and the inhibition of platelet aggregation are aspects that contribute to the cardioprotective effects of olive tree leaves [109]. As reported above, OL has activity on calcium channels; this evidence opens up new perspectives for using this molecule in many pathologies and also neurodegenerative diseases, such as Alzheimer’s disease (AD). In fact, neurodegenerative pathologies are often caused by calcium cytotoxicity, and in this case, the olive’s polyphenols, such as oleuropein, could play an important role.

Several studies have already been performed in this direction, and the results are encouraging. Transgenic mice (APPswe/PS1dE9) received, from 7 to 23 weeks of age, 50 mg/kg of oleuropein contained in OLE compared to a control diet [110]. OLE-treated mice showed significantly reduced (*p* < 0.001) amyloid plaque deposition in cortex and hippocampus compared to control mice, providing a basis for considering natural and low cost biophenols from olive as a promising drug candidate against AD. Nevertheless, additional studies are needed to validate these results and determine the anti-amyloid mechanism, bioavailability as well as permeability of olive biophenols to the blood brain barrier (BBB) in AD.

Oxidative stress certainly contributes to the onset of neurodegenerative diseases. OLE, in combination with hibiscus flower extract, has been shown to prevent the degeneration of cerebral cells following insult in vitro studies, thus exerting a neuroprotective action [111]. The action of the pool of molecules that these extracts contain is related to the bioavailability often impaired by oral administration. With a “drug like” approach, the components of the phytocomplexes were tested for their ability to permeate the BBB using an in silico predictive model. Oleuropein, contained in the mix, was shown to be able to pass the BBB and, using adequate doses of leaf extract, it was possible to reach biologically active concentrations in the brain, demonstrating the neuroprotective efficacy in the brain and its permanence even after oral intake, as confirmed by in vivo studies [63].

The anti-inflammatory effect of OLE has been investigated by screening all diseases in which inflammatory mechanisms are involved. In order to demonstrate the anti-inflammatory properties of OLE on upper respiratory illness (URI), very common among teenagers and especially in young athletes, a study was performed on high school students by treating for nine weeks the groups either with 100 mg of oleuropein or with placebo [112]. The young athletes were monitored during training, and the illness incidence was the same in both groups, but the treatment with OL led to a reduction of the sick days, resulting in a quicker recovery.

OLE also plays an important role against osteoarthritis (OA) [113]. A study revealed as an olive oil supplemented diet could improve cartilage recovery after anterior cruciate ligament transection [114]. In particular, the polyphenols inhibited the development of proinflammatory cytokines, including IL-1β, TNF-α, IL-6, and prostaglandin E_2_, and other synthetic pathways involved in the development and progression of OA [115,116].

Table 4 summarizes the principal biological activities of OLE reported in the literature.

### 3.6. Cosmetic Formulations of Olive Leaf Extracts

Cosmetic products currently on the market often contain “*Olea europaea* Leaf Extract”, whose composition includes the presence of TY, HT, OL, and other flavonoids, such as luteolin and apigenin. Some studies attested the strong antioxidant activity of OLE on the skin; for example, oleuropein formulations highlighted lenitive efficacy by reducing erythema, transepidermal water loss, and blood flow of about 22%, 35%, and 30%, respectively [117].

The rejuvenating effect of OLE in cosmetics was also studied on 36 people who used a particular cream containing the extract “SUPERHEAL™ O-Live Cream” (PhytoCeuticals, Inc, USA) [118]. After two months of daily applications, OLE led to an amelioration in overall skin condition concerning hydration, wrinkle state, and erythema conditions, as determined by measuring several physiological parameters, such as melanin and erythema index, transepidermal water loss, skin hydration, skin pH, sebum level, texture, and wrinkles.

Another important aspect that can be considered about a cosmetic activity is the photoprotective effect of polyphenols. This potential effect has been studied in oral and topical photoprotection. There is a lot of interest in researching natural sunscreen, also considering the low impact on the environment. An in vitro assay on sun protection factor (SPF) and molecular model studies of UV absorption supported the use of OLE as a photoprotective, antioxidant, and antimutagenic agent.

Skin cancer is one of the most common types of cancer, and it is becoming more impactful day by day. In this regard, the scientific world is trying to find a valid way of prevention that can fight even less severe reactions from sun exposure, such as erythema, photoaging, and immunosuppression [119,120]. Finding a 100% preventive photoprotective filter from natural sources—and also from waste—could be a great starting point for the development of some preventive products able to reduce the frequency of this type of chronic disease.

### 3.7. Secondary Metabolites and Biological Activity of Kernel and Seed Extracts

Olive stone is a lignocellulosic material, with hemicellulose, cellulose, and lignin that are the main components. Olive stones are obtained by separation of the pulp from the kernels by means of two different technologies, both before the EVOO extraction process (through the employment of the destoner [121] separating the whole kernel from the fruit) and after the EVOO extraction process (through olive pomace depicting machine). The utilization of the lignocellulosic material from olive stones in biofuel production has been recently reported [122].

Alu’datt et al. optimized various extraction conditions and characterized the phenolic olive seed compounds as well as their antioxidant activity [123]. Their research revealed that the free phenolic forms were predominant in olive seeds. In 1998 Fernández-Bolaños et al. analyzed both the water-soluble non-carbohydrate compounds obtained by steam explosion, such as sugar degradation compounds (furfural and hydroxymethylfurfural), lignin degradation compounds (vanillic acid, syringic acid, vanillin, and syringaldehyde), and phenolic olive fruit compounds (TY and HT) [124]. As a result, they observed that the concentration of hydroxytyrosol was higher than that of the other compounds. In addition, they noted that the amount of HT increased by raising both steaming temperature and time. Rodríguez et al. later confirmed olive stone as an attractive source of bioactive and valuable compounds due to the presence of polyphenols and polyols [125]. They also explored various potential uses of this EVOO by-product, such as activated carbon, furfural production, plastic filled, abrasive, cosmetics, biosorbents, animal feed, and resin, discussing the application of this material based on each component.

González-Hidalgo et al. analyzed the composition of TY, HT, OL, and tocopherol and the antioxidant activity in different fractions of the main by-product from the table olive canning industry (i.e., the stone with some residual olive flesh) [126]. The highest polyphenolic concentration (1710.0 ± 33.8 mg/kg), as well as the highest antioxidant activity (8226.9 ± 9.9 hydroxytyrosol equivalents mg/kg), were observed in the seed olive. The highest amounts of HT (854.8 ± 66.0 mg/kg) and TY (423.6 ± 56.9 mg/kg) were registered in the whole by-product from the pepper stuffed olives, while the maximum OL content (750.2 ± 85.3 mg/kg) was reported in the stone without seed. In particular, α-tocopherol values of 79.8 ± 20.8 mg/kg and 6.2 ± 1.2 mg/kg were registered in the seed olive stone and in the whole by-product from the anchovy-stuffed olives, respectively. In light of these results, the use of table olive by-product could be a source of natural antioxidants in food, cosmetic, or pharmaceutical products. In addition, table olive by-product revaluation could help to diminish their environmental impact.

Recently, Sibel Bolek proposed to replace wheat flour with olive stone powder, rich in fiber and antioxidants derivatives, in biscuit production, to explore its effect on the rheological characteristics and quality of dough [127]. They added 0%, 5%, 10%, and 15% of olive stone powder in place of the same amounts of wheat flour. As a result, wheat flour replacement with olive stone powder increased the antioxidant activity, as well as fat and fiber content of sample biscuits. In particular, 30.44% ± 0.03% DPPH radical scavenging activity, 11.22% ± 0.09% crude fiber, and 26.32% ± 0.22% fat were quantified by substituting wheat flour with 15% olive stone powder. Furthermore, the authors showed that a replacement of wheat flour with up to 15% olive stone powder did not cause any alteration to the biscuit sensorial properties.

However, olive fruits present large variability in composition. Khadem et al. investigated the physicochemical properties and bioactive contents of whole olive stone oils extracted from six olive varieties, namely Zard, Roughani, Mari, Shengeh, Koroneiki, and Manzanilla, cultivated in the city of Fasa, Iran [128]. They analyzed fatty acids, sterols, and triacylglycerols contents, equivalent carbon number, saponification, iodine, unsaponifiable matter values, and phenolic contents, concluding that, despite the great variability in the whole olive stone oils composition among the six cultivars, whole olive stone oils could be used as a natural source of polyphenol compounds for human consumption.

Lama-Muñoz et al. proposed a multi-step process that could allow an integral use of olive stone from the point of view of a biorefinery plant [129]. They proposed an initial aqueous extraction at 130 °C for 90 min without acid addition and a solid:liquid ratio of 1:2 (*w/w*), useful to recover liquors with higher phenolic content and antioxidant capacity. This first step provided a double benefit: to separate phenolic compounds potentially useful in cosmetic, pharmaceutical, and food industries and to recover biomass of possible inhibitors. In a second time, olive stones were exposed to further treatment with 2% (*w/v*) sulfuric acid to obtain the maximum amount of fermentable sugars, mainly xylose, with a low content of compounds such as formic acid, furfural, and hydroxymethylfurfural, able to inhibit fermentative microorganisms involved in bioethanol production. The remaining olive stone was particularly rich in cellulose and lignin, and it could be subjected to enzymatic hydrolysis to achieve glucose in high yields. Glucose could be then converted into bioethanol or into other products, such as poly(3-hydroxybutyrate) and hydroxymethylfurfural. The final lignin-enriched solid could be converted into phenols, biopolymers, or fibers or directly used for energy production. The authors thus concluded that olive stone might be considered as an excellent feedstock for biorefinery plant development. A review written by Ruiz et al. in 2017 described the most recent proposals for the use of biomass derived from olive tree cultivation and olive oil production processes [130].

Spizzirri et al. obtained an ethanolic extract with antioxidant properties to be used in the food and cosmetic industry as a functional food and nutraceutical additive, starting from the olive stones discarded from the EVOO production [131]. The efficiency of the multi-step extraction method was evaluated by quantifying the recovery yield and the total phenolic compounds for a series of solvents with different polarities. Flavonoids were shown to represent about 60% of phenolic antioxidants. The antioxidant activity of the alcoholic fraction was then determined by DPPH assay, showing a good efficiency already at low concentration (IC_30_ of 0.060 mg/mL). In addition, the extract showed an interesting ability to preserve β-carotene from lipidic peroxidation (IC_30_ of 1.30 mg/mL).

## 4. *Olea europea* L. By-Products for Zootechnical Feeding

In addition to their use as health-promoting compounds, by-products of the olive oil industry could also represent a source of ingredients for zootechnical feeding. Feeding innovations based on the utilization of these bioactive-rich by-products can reduce enteric emissions in ruminants while improving the nutritional composition and shelf-life quality of meat and meat products, simultaneously improving environmental sustainability [132]. The by-products of olive oil production, which represent an important environmental issue in the Mediterranean area, can be valorized in the livestock sector according to the ‘pyramid of the value of the bioeconomy’, which favors the use of functional ingredients of high value for animal nutrition. In this regard, the content of poly-unsaturated fatty acids is improved in oils for human consumption, while saturated fatty acids are employed for animal feeding. This makes the food healthier for humans while simultaneously reducing feeding costs and the environmental impact of livestock [133,134].

Furthermore, the animal diet deeply influences the quality of the animal meat and derived products and, consequently, the quality of the human diet and health. An animal diet enriched with polyphenols olive oil waste could represent a sustainable approach both for reducing adverse environmental effects of these wastes and for improving the quality of the products of animal origin.

It is important to underline that the use of olive pomace, containing appreciable amounts of oil, has already been considered a feasible strategy to influence the quality of meat [135]. OMW has also been exploited for animal feeding. Gerasopoulos et al. separated, by means of a microfiltration method, the two liquid products from olive mill wastewater, i.e., the downstream permeate and the upstream retentate, and, after characterization, incorporated them into broilers’ feed [136]. By measuring oxidative stress biomarkers in blood and tissues, they noted that broilers given OMW-supplemented feed had significantly lower levels of protein oxidation and lipid peroxidation and higher total antioxidant capacity in plasma and tissues compared to the control group. As already known, an antioxidant status able to reduce the stress level in broiler chickens could improve meat quality [137].

In order to improve growth performance and feed digestibility of pigs and pork meat quality, Paiva-Martins et al. investigated the supplementation of animal feeds with olive leaves [138]. Unfortunately, they observed that pigs fed diets with olive leaves showed a lower daily weight gain and a decrease in overall backfat compared to pigs fed by the conventional diet. However, chops from pigs fed the leaf diets had lower peroxide and conjugated diene contents, a lower drip loss, and an improved oxidative stability thanks to a significantly higher α-tocopherol concentration in intramuscular fat and backfat.

Milk quality is also affected by the introduction of by-products from the olive oil industry in animal feeding. Arco-Pérez et al. assessed the effect of the partial replacement of the forage in the diet with olive by-products in goats feeding obtaining milk with higher amounts of vaccenic, eicosadienoic, and conjugated linoleic acid, valuable molecules with several beneficial effects, with the concomitant improvement of the animal meat quality [139]. On the other hand, Branciari et al. investigated both the nutraceutical profile and quality characteristics of the cheese deriving from sheep feed with an OMW-enriched diet [140]. The polyphenol supplementation yielded TY and HT sulfate metabolites both in the obtained milk and cheese derivatives, also providing a direct antioxidant effect on cheese without modifying its chemical composition.

Kerasioti et al. studied the tissue specific effects of feeds supplemented with OMW on detoxification enzymes in sheep, which resulted in an increased glutathione *S*-transferase activity in the liver and spleen and a decreased γ-glutamylcysteine synthetase expression in the liver, without affecting the superoxide dismutase activity in both tissues [141]. The authors concluded that the beneficial effects of the OMW-enriched feeds were tissue- and developmental stage-specific. Instead of using OMW, Musawi et al. conducted a study to investigate the effect of ground olive leaves supplementation on milk yield and composition, as well as on some blood biochemical parameters, in goats [142]. Although the diet had no significant effect on the average animal body weight, the milk production was significantly increased in goats fed with 2% olive leaves powder. Nevertheless, milk compositions (lactose, protein, and fat percentage) and energy value, as well as blood and biochemical parameters of the ruminant, did not vary significantly from the control group.

Similar to milk and cheese, egg quality is affected by the introduction of by-products of the olive oil industry in animal feeding. Zangeneh and Torki evaluated the performance of laying hens fed with olive pulp [143]. Although the olive pulp-included diet had no significant effect on overall egg production and mass, eggshell weight was higher than that of the birds fed with the control diet, suggesting no deleterious effects on bird’s performance but yielding more resistant eggs. Cayan and Erener also conducted an experiment aimed at measuring the effects of olive leaves powder on performance, egg yield, egg quality, and yolk cholesterol level of laying hens [144]. In this case, the authors noted that the supplementation had no effect on feed intake and egg weight and yield, but it significantly increased the final body weight of hens. Furthermore, the dietary olive leaves powder increased yellowness in yolk color and decreased its cholesterol content by about 10%.

## 5. Conclusions

It is well-known that EVOO and the by-products of its production are an important source of bioactive compounds, e.g., polyphenols and other secondary metabolites, that contribute to reducing cellular oxidative stress and inflammation, thus potentially supporting the resolution of many pathologies. Although the beneficial effects of EVOOs are recognized, e.g., in the Mediterranean Diet, currently, olive by-products are not yet properly exploited, except for a few cosmetic formulations on the market. In fact, there are many regulatory obstacles that prevent these by-products, still considered waste, from re-entering the food or nutraceutical formulations sector. Nevertheless, both olive mill wastewater and olive leaves extracts have already demonstrated peculiar properties. Similarly, kernel and seed extracts were shown to have great potential as nutraceuticals and cosmeceuticals. In addition, besides the direct exploitation for human purposes, all these by-products could be easily employed in animal feeding, thus positively affecting the quality of products for human consumption, e.g., milk, cheese, eggs, and meat. As such, their exploitation would benefit both our health and that of the environment by reducing their waste disposal. With the aim of favoring the transition from a linear to a circular economy within the olive mills, a revision of the legislation, an improvement of the environmental governance, and the identification of economic tools is needed for creating adequate incentives for adopting circular and sustainable production and consumption models, and also promoting the transition towards environmental tax reform. Although this project is quite ambitious and time-consuming, we hope that research studies devoted to demonstrating its feasibility, similarly to those reported in this review article, will contribute to its realization.

## Figures and Tables

**Figure 1 molecules-26-01072-f001:**
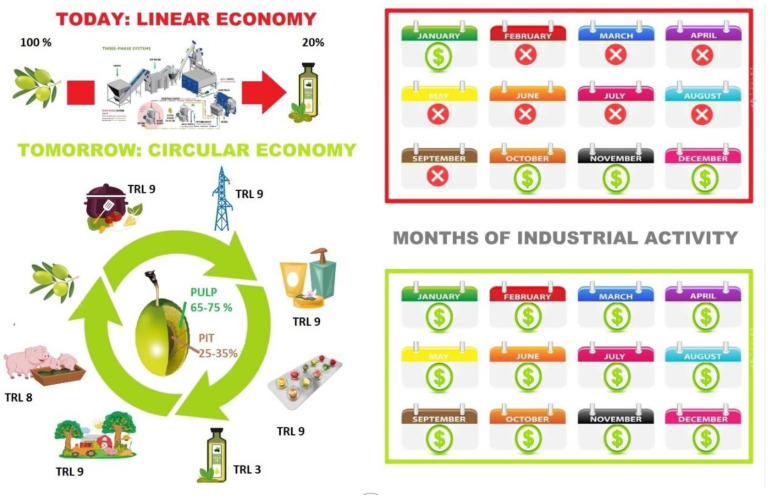
From linear to the circular economy in the olive oil sector.

**Figure 2 molecules-26-01072-f002:**
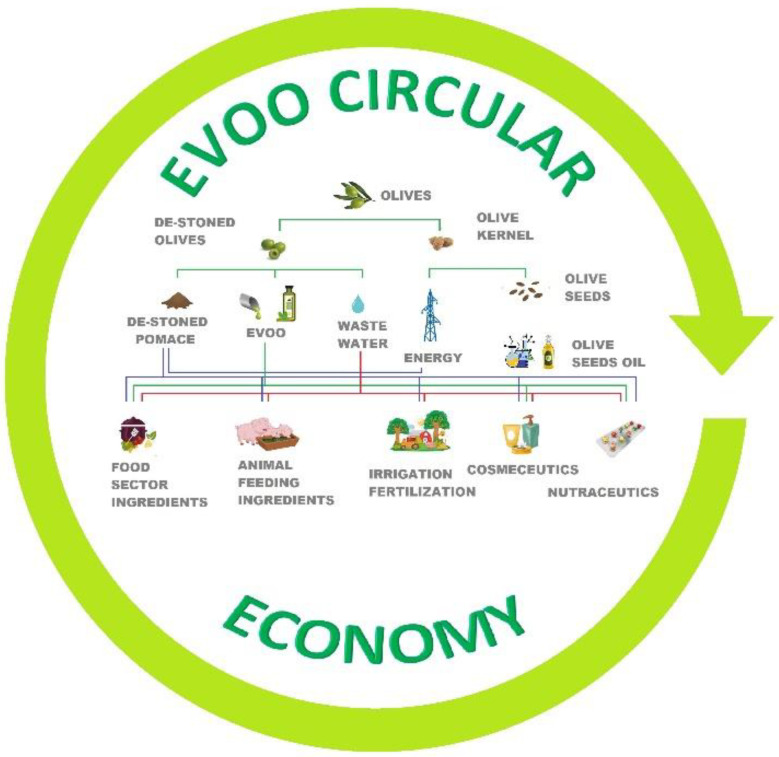
Extra virgin olive oil’s (EVOO’s) circular economy: overview of an integrated olive tree exploitation.

**Figure 3 molecules-26-01072-f003:**
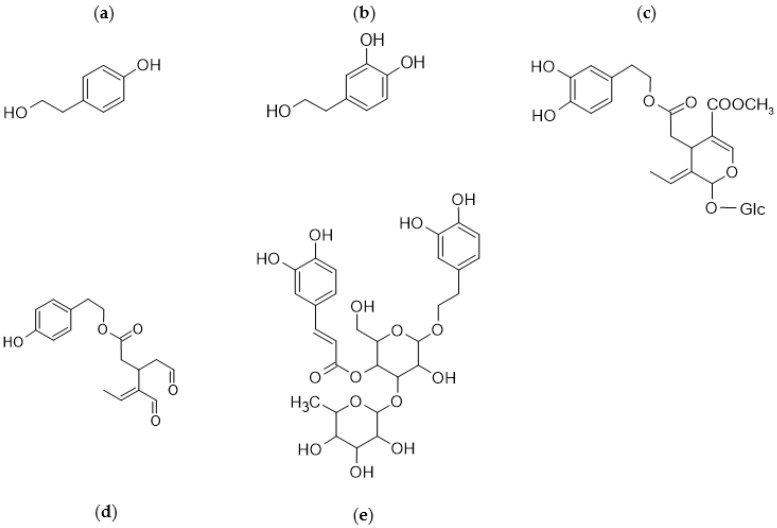
Most abundant polyphenols in olives: (**a**) tyrosol; (**b**) hydroxytyrosol; (**c**) oleuropein; (**d**) oleocanthal; (**e**) verbascoside.

**Figure 4 molecules-26-01072-f004:**
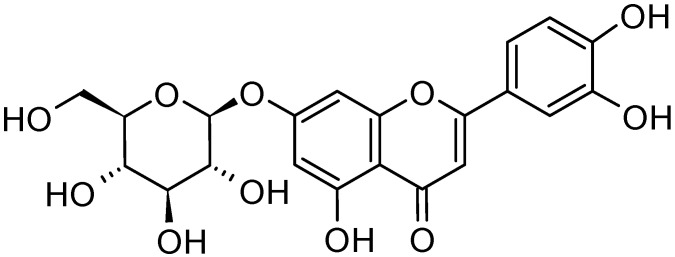
Chemical structure of luteolin 7-*O*-glucoside.

**Table 1 molecules-26-01072-t001:** Distribution of the main classes of metabolites in the different parts of the plant *Olea europea* L. [13,14,15,16,17,18,19,20,21].

Seed Oil	Virgin Olive Oil	Skin	Pulp	Wood	Leaves
Phenolic acid/aldehydes	Phenolic acid/aldehydes	Phenolic acid/aldehydes	Phenolic acid/aldehydes	Phenolic acid/aldehydes	Phenolic acid/aldehydes
Tocopherols	Tocopherols		Tocopherols		
Sterols	Sterols	Organic acid and coumarins	Organic acid and coumarins	Organic acid and coumarins	Organic acid and coumarins
		Flavonoids	Simple phenols and derivatives	Simple phenols and derivatives	Simple phenols and derivatives
		Lignans	Secoiridoids and derivatives	Secoiridoids and derivatives	Secoiridoids and derivatives
		Fatty acids and derivatives		Flavonoids	Flavonoids
		Pentacyclic triterpenes			Tocopherols

**Table 2 molecules-26-01072-t002:** Chemical composition of olive mill wastewater (OMW) extracts obtained by different techniques.

Composition	OMW ^a^	UF ^b^	AIR ^c^	WSF ^c^	WIF ^c^	UF HSF			UF ETNA O1PP			NF 90	
						F ^a^	P ^a^	R ^a^	F ^a^	P ^a^	R ^a^	F ^a^	R ^a^
Total phenols	1409.0	1692.0	3.2	2.8	1.9	81.3	79.5	81.3	75.5	62.2	77.4	65.6	86.2
Hydroxytyrosol	3.8	n.d.	-	-	-	3.8	3.7	3.9	3.5	3.0	3.8	3.2	4.0
Protocatechuic acid	25.0	-	-	-	-	25.0	24.0	24.5	27.0	20.6	26.0	22.0	30.0
Catechol	7.5	-	-	-	-	7.5	7.1	7.2	6.0	5.0	6.2	5.5	7.5
Tyrosol	39.0	n.d.	-	-	-	39.0	38.7	39.6	34.2	30.0	36.0	31.0	40.0
Caffeic acid	5.0	-	-	-	-	5.0	4.9	5.2	4.0	3.0	4.4	3.2	3.7
*p*-Cumaric acid	1.0	-	-	-	-	1.0	0.9	0.9	0.8	0.6	1.0	0.7	1.0
Verbascoside	-	n.d.	-	-	-	-	-	-	-	-	-	-	-
Isoverbascoside	-	n.d.	-	-	-	-	-	-	-	-	-	-	-
Carbohydrates	-	-	25.0 ^d^	60.0 ^d^	5.1 ^d^	-	-	-	-	-	-	-	-
Fucose	-	-	0.5	0.6	0.4	-	-	-	-	-	-	-	-
Rhamnose	-	-	14.3	13.7	16.4	-	-	-	-	-	-	-	-
Arabinose	-	-	14.1	10.7	17.6	-	-	-	-	-	-	-	-
Galactose	-	-	12.6	13.1	5.9	-	-	-	-	-	-	-	-
Glucose	-	-	42.2	45.1	47.7	-	-	-	-	-	-	-	-
Mannose	-	-	5.5	5.4	4.4	-	-	-	-	-	-	-	-
Xylose	-	-	5.0	4.9	5.3	-	-	-	-	-	-	-	-
Galacturonic acid	-	-	4.9	5.0	1.4	-	-	-	-	-	-	-	-
Glucuronic acid	-	-	1.0	1.1	0.7	-	-	-	-	-	-	-	-
Proteins	-	-	3.2	11.0	0.3	-	-	-	-	-	-	-	-

UF = ultrafiltration residue; AIR = alcohol insoluble OMW residue; WSF = water-soluble fraction; WIF = water-insoluble fraction; UF HSF = ultrafiltration performed with HSF membrane type; UF ETNA O1PP = ultrafiltration performed with ETNA 01PP membrane type; NF 90 = nanofiltration performed with NF90 membrane type; F = feed; P = permeate; R = retentate. ^a^ Expressed as mg/L; ^b^ expressed as ppm; ^c^ expressed as g/100 g of fraction; ^d^ expressed as mol%; n.d. = not determined.

**Table 3 molecules-26-01072-t003:** Total phenols contents and antioxidant activities of different cultivars of olive leaves extracts.

Cultivars	Total Phenol ^1^	FRAP ^2^	DPPH ^3^
Manzanilla	134.50 ± 0.01	1107.71 ± 0.01	33.93
Conservolea	92.35 ± 0.01	1277.33 ± 0.01	62.94
Arbequina	42.35 ± 0.02	1760.57 ± 0.01	62.56
Mishen	71.93 ± 0.01	1971.37 ± 0.01	63.48
Coratina	155.91 ± 0.06	358.66 ± 0.01	22.95
Roghani	121.75 ± 0.02	1400.76 ± 0.01	29.58
Kalamon	190.65 ± 0.03	532.76 ± 0.01	26.74
Amphissis	50.70 ± 0.01	1110.38 ± 0.01	95.39
Yellow	73.85 ± 0.01	1400.95 ± 0.01	53.80
Amigdalolia	42.73 ± 0.01	1341.05 ± 0.01	74.30
Mary	62.24 ± 0.01	1203.81 ± 0.01	60.26
Leccino	59.23 ± 0.01	568.28 ± 0.01	69.30
Shenge	61.97 ± 0.01	614.19 ± 0.01	60.18
Gordal	184.72 ± 0.01	450.86 ± 0.01	20.66
Sevillenca	83.63 ± 0.01	432.19 ± 0.01	34.92
Fishomi	109.98 ± 0.06	1794.57 ± 0.01	32.82

^1^ Expressed as mg GAE/g dry extract. ^2^ Expressed as µmol Fe II/g dried extract. ^3^ Concentration expressed in IC_50_: µg/mL.

**Table 4 molecules-26-01072-t004:** Biological effects of olive leaf extracts (OLEs).

Disease	Type of Experiment	Dose	Effects
Hypertension [100]	Human clinical trial	1000 mg OLE/die	Lowering systolic and diastolic blood pressures, significant reduction of triglyceride (TG) levels.
Atherosclerosis [107]	in vivo	100 mg OLE/kg body weight	Reduction of the levels of cholesterol, TGs, and LDL cholesterol, and block of the inflammatory response.
Thrombosis [109]	in vitro	1% *v*/*v* OLE	Significant dose-dependent reduction in platelet activity.
Hypocholesterolemia [111]	Human studies	1.2 g OLE/die	Reduction of total cholesterol, decreased LDL cholesterol.
Diabetes [106,108]	in vitro	IC_50_ = 4.0–0.02 mg/mL OLE	Inhibition of the activities of α-amylases from human saliva and pancreas.
	Human clinical trial	500 mg OLE/die	Significant reduction in HbA1C values.
Alzheimer [110]	in vivo	50 mg OLE/kg	Reduction of amyloid plaque deposition in cortex and hippocampus.
Upper respiratory illness [112]	Randomized controlled trials	100 mg oleuropein/die	Reduction of the sick days, i.e., acceleration of the recovery.

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
