# Peer review of "Olive Tree in Circular Economy as a Source of Secondary Metabolites Active for Human and Animal Health Beyond Oxidative Stress and Inflammation"

_molecules, 2021, doi:10.3390/molecules26041072_

Round 1
Reviewer 1 Report
This literature review presents an overview of the biological activities that these by-products have been shown to possess and the main formulations currently on the market. However, the paper appears to have numerous weaknesses such many sentences were verbatim plagiarism. The manuscript needs some revisions to improve its quality.
- In this text, many sentences were verbatim plagiarism from references. Those sentence should be rewritten to make sure the sentence follows the grammatical rules.
- L385-395, 540-548 , 685-689, 715-720 from J. Mol. Sci.2016, 17(12), 2042; https://doi.org/10.3390/ijms17122042.
- L 443-448 from J. Agric. Food Chem.2006, 54, 2, 434–440.
- 369-381 from Chem. Eng. Trans. 2016, 49. DOI: 10.3303/CET1649080
- 586-593 from https://doi.org/10.1007/s11746-998-0106-8.
- 601-609 from https://doi.org/10.1111/j.1365-2621.2011.02892.x
- 612-620 from Innov. Food Sci. Emerg. Techno, 2020, 64, 102423.
- L633-642 from Industrial Crops Prod. 2014, 60, 30-38.
- L177-181, 408-409, Sep. Purif. Techno, 2017, 172, 310-317.
- ……...etc.
- Please provide the full name of EVOO and EFSA in Abstract section.
- Scientific name “Olea europaea” must to write correctly and revise the whole manuscriptcarefully.
- There were many errors throughout the paper therefore the paper should be carefully checked. Ex:
- Line 36, Change “Olive mill waste water (OMM)” to “Olive mill wastewater (OMM)”.
- Line 112, Change “ Olea Europea L.” to “2. Olea europea L.”
- Line 116, Change “Table 1. Main chemical compound” to “Table 1. Main chemical compounds”
- Line 396, Change “3.4. Olive leaves Extract” to “3.4. Olive leaves extract”.
- Line 552, Change “3.6. Olive leaf Cosmetic formulation” to “3.6. Olive leaf cosmetic formulation”
- Line 575, Change “3.7 Kernel and Seed biological activity” to “3.7 Kernel and seed biological activity”.
- This references did not follow this journal format and uniform.
According to the above concerns, this article should not be accepted for publication.
Author Response
Answer to Comments and Suggestions for Authors
Reviewer 1
Comment: This literature review presents an overview of the biological activities that these by-products have been shown to possess and the main formulations currently on the market. However, the paper appears to have numerous weaknesses such many sentences were verbatim plagiarism. The manuscript needs some revisions to improve its quality.
Answer:
We thank the reviewer for the comment. The work has been revised in its entirety and many data, already subject to previous publications relating to the effects of the individual components present in the oil, have been eliminated by preserving the bibliographic references (integrated with more recent ones) more relevant to the topic and by eliminating those that were not congruent. Extensive editing of the English language and style has been performed and the plagiarized sentences eliminated.
Comment: In this text, many sentences were verbatim plagiarism from references. Those sentence should be rewritten to make sure the sentence follows the grammatical rules.
- L385-395, 540-548, 685-689, 715-720 from J. Mol. Sci.2016, 17(12), 2042; https://doi.org/10.3390/ijms17122042.
- L 443-448 from J. Agric. Food Chem.2006, 54, 2, 434–440.
- 369-381 from Chem. Eng. Trans. 2016, 49. DOI: 10.3303/CET1649080
- 586-593 from https://doi.org/10.1007/s11746-998-0106-8.
- 601-609 from https://doi.org/10.1111/j.1365-2621.2011.02892.x
- 612-620 from Innov. Food Sci. Emerg. Techno, 2020, 64, 102423.
- L633-642 from Industrial Crops Prod. 2014, 60, 30-38.
- L177-181, 408-409, Sep. Purif. Techno, 2017, 172, 310-317.
……...etc.
Answer:
We perfectly agree and thank the reviewer for the suggestion. As already stated, the entire article has been revised checking for plagiarism. A deep English and editing revision has been also performed and the changes have been highlighted in red in the version with the track changes activated (provided to the referees). We hope that this revised version is now easier to read and error-free.
Comment: 2. Please provide the full name of EVOO and EFSA in Abstract section.
Answer:
The full name of EVOO and EFSA in Abstract section have been provided.
Comment 3. Scientific name “Olea europaea” must to write correctly and revise the whole manuscript carefully.
Answer:
The scientific name “Olea europea L.“ has been correctly written in each section of the paper.
Comment 4. There were many errors throughout the paper therefore the paper should be carefully checked. Ex:
- Line 36, Change “Olive mill waste water (OMM)” to “Olive mill wastewater (OMM)”.
- Line 112, Change “Olea Europea L.” to “Olea europea L.”
- Line 116, Change “Table 1. Main chemical compound” to “Table 1. Main chemical compounds”
- Line 396, Change “3.4. Olive leaves Extract” to “3.4. Olive leaves extract”.
- Line 552, Change “3.6. Olive leaf Cosmetic formulation” to “3.6. Olive leaf cosmetic formulation”
- Line 575, Change “3.7 Kernel and Seed biological activity” to “3.7 Kernel and seed biological activity”.
- This references did not follow this journal format and uniform.
Answer:
As already stated, the manuscript has been carefully revised following the suggestions above. All references have been formatted following journal guidelines.

Reviewer 2 Report
The manuscript “Olive tree in circular economy: a source of secondary 2 metabolites active for human and animal health: not 3 only against oxidative stress and inflammation” submitted to Molecules by Dr. Mallamaci and co-workers presents an interesting overview of the composition and biological activities that the main by-products of the olive oil industry such as olive wastewater, olive pomace, olive leaves and olive stone have been shown to possess and the main formulations currently on the market to use these by-products to make a circular economy around olives an important element of the Mediterranean landscape. Obviously, this review is interesting but needs a profound review in my opinion.
Line 42. Olive oil is a characteristic (more than important) element of the Mediterranean diet.
Table 1 should be revised considering that class chemical should be paired to the first compound of this class. How were ordered compounds into a class? Compounds appear without apparent order. I suggest order these compounds considering the quantitative presence in olive oil. This last change will improve the quality of the Table.
Line 126-128. The review says “The secondary metabolites from Olea Europea L. have high biological value and they are present in different concentrations in the different parts of the olive plant (Table 2); as such, many of them are also present in the derived EVOOs but they can be also found in the resulting waste products”. Table 2 can include concentration information as the text suggested. A new Table with the main compounds presents in wastewater and other by-products is necessary considering the context of the review.
Line 138. The text says, “that can be defined as nutraceuticals for their nutritional and sensory characteristics” In my opinion polyphenols can be defined as nutraceuticals for their biological/pharmacological actions (see doi: 10.1016/j.bcp.2018.07.050).
Abbreviations should be presented the first time. What means OL? (line 142) What means OMW? (line 158). What means HT? (line 211)
Line 165-169. The nutritional and health-promoting effects of olives and olive oils should be presented using reviews (see doi: 10.1016/j.bcp.2012.07.017) and suitable clinical studies (see doi: 10.1016/j.clnu.2012.08.002). I believe that the review has too much in vitro and in vivo information of biological actions of olive oil components but few clinical information.
Line 211-218. What means HT derivatives? LDH is a cytotoxic biomarker but not an inflammatory biomarker. Please confirm that “ex vivo” is correct.
Line 236. I cannot understand some sentences. Here, what means “the cytoprotective effects of formulations containing both syntethic HT and that derived from OMW…”? What means synthetic HT? Syntethic should be changed by synthetic.
The manuscript presents a lot of typographical mistakes that should be corrected.
Line 258. “The use of OMW phenolic compounds in milk-based beverages has also been reported to improve their nutritional properties” What happens with their organoleptic qualities?
Line 366-384. This paragraph is too much extent (in a similar way to other paragraphs along the review) and should included clinical antecedents about the topic olive oil consumption and hepatic steatosis (see doi: 10.1093/jn/nxz147).
Line 409. “There is a growing interest in extracting and separating oleuropein from olive leaves because natural active compounds are safer for human health than synthetic chemicals”. I cannot understand this sentence. Why natural phenols are safer than synthetic phenols?
Line 460-474. As I previously mentioned clinical studies should be included to improve the description of the healthy actions of olive oil/olive oil components. Here, the antihypertensive actions of olive components should be complemented with clinical studies (i.e. doi: 10.1007/s00394-015-1060-5).
Line 533. What means “human body districts”?
Some parts of the review are too much descriptive (i.e. line 598-609). I believe that reviewed those will improve the quality of the manuscript.
The title of the sections should be also revised. In my opinion, titles are not appropriate in same parts of the manuscript (i.e. “Kernel and seed biological activity”).
Along the manuscript emerge the idea that the mix compounds in by-products can be more effective than a only compound considering the possibility of an additive/synergistic effects of bioactive olive compounds as have been reported (i.e. doi: 10.1021/acs.jafc.9b04816). Perhaps this idea should be directly presented in the context of the topic.
In my opinion the review needs additional figures or revised the figures (1, 2) to improve the representation of the by-products of the olive oil industry, the quantities produced, the contamination, the different composition of each by-product, the main alternative applications of each by-product,….
Author Response
Answer to Comments and Suggestions for Authors
Reviewer 2
Comment: The manuscript “Olive tree in circular economy: a source of secondary 2 metabolites active for human and animal health: not 3 only against oxidative stress and inflammation” submitted to Molecules by Dr. Mallamaci and co-workers presents an interesting overview of the composition and biological activities that the main by-products of the olive oil industry such as olive wastewater, olive pomace, olive leaves and olive stone have been shown to possess and the main formulations currently on the market to use these by-products to make a circular economy around olives an important element of the Mediterranean landscape. Obviously, this review is interesting but needs a profound review in my opinion.
Answer:
We thank the reviewer for the comment. The manuscript has been revised in each section by focusing our attention on the topic of the paper as well as on the figures and tables, also according to editor suggestions.
Extensive editing of the English language and style has been performed and the plagiarized sentences have been eliminated.
The article now contains an adequate number of citations, including those indicated by the reviewer.
Comment 1: Line 42. Olive oil is a characteristic (more than important) element of the Mediterranean diet.
Answer:
On L42 “important” has been substituted with “a characteristic”
Comment 2: Table 1 should be revised considering that class chemical should be paired to the first compound of this class. How were ordered compounds into a class? Compounds appear without apparent order. I suggest order these compounds considering the quantitative presence in olive oil. This last change will improve the quality of the Table.
Answer:
Table 1 has been removed from the original paper in according to editor suggestion: “there is no need for such a detailed table, all compounds present in the table cannot be recycled or are found in variables quantities.”
Comment 3: Line 126-128. The review says “The secondary metabolites from Olea Europea L. have high biological value and they are present in different concentrations in the different parts of the olive plant (Table 2); as such, many of them are also present in the derived EVOOs but they can be also found in the resulting waste products”. Table 2 can include concentration information as the text suggested. A new Table with the main compounds presents in wastewater and other by-products is necessary considering the context of the review.
Answer:
Distribution of the main classes of metabolites in the different parts of the plant Olea europeae L. (Table 1) is precisely reported in the text from reference 12 to reference 19 of the original paper.
For this reason, L129 “Table 2. Distribution of the main classes of metabolites in the different parts of the plant Olea europeae L.” of the original paper becomes as follows “Table 1. Distribution of the main classes of metabolites in the different parts of the plant Olea europeae L.[13-21]”. Consequently, the numbering of the tables has been changed.
The main compounds in olive wastewater as well as the other by-products are now reported in the Table 2.
Comment 4: Line 138. The text says, “that can be defined as nutraceuticals for their nutritional and sensory characteristics” In my opinion polyphenols can be defined as nutraceuticals for their biological/pharmacological actions (see doi: 10.1016/j.bcp.2018.07.050).
Answer:
The phrase on L138 “can be defined as nutraceuticals for their nutritional and sensory characteristics” of the original paper has been rewritten as follows “can be defined as nutraceuticals for their biological/pharmacological actions” important”.
Reference [13] “Beltran, G.; Aguilera, M.P.; Del Rio, C.; Sanchez, S.; Martinez, L. Influence of fruit ripening process on the natural antioxidant concentration of Hojiblanca virgin olive oils. Food Chem. 2005, 89, 207–215.” of the original paper has been replaced as follows [15] “Tresserra-Rimbau, A.; Lamuela-Raventos, R. M.; Moreno, J.J. Polyphenols, food and pharma. Current knowledge and directions for future research Biochemical Pharmacol. 2018, 156, 186–195.”
Comment 5: Line 165-169. The nutritional and health-promoting effects of olives and olive oils should be presented using reviews (see doi: 10.1016/j.bcp.2012.07.017) and suitable clinical studies (see doi: 10.1016/j.clnu.2012.08.002). I believe that the review has too much in vitro and in vivo information of biological actions of olive oil components but few clinical information.
Answer:
The sentence “To candidate olive oil components as responsible of the benefits of the MD, Fernandes et al. summarized evidences from randomized controlled trials on the effect of regular dietary intake of olive oil on inflammatory markers [34]” has been written to support clinical information about beneficial effects of olive oil.
In addition, references [21] and [31] have been respectively substituted by reference [23] (doi: 10.1016/j.bcp.2012.07.017) and reference [33] (doi: 10.1016/j.clnu.2012.08.002), as suggested.
Comment 6: Abbreviations should be presented the first time. What means OL? (line 142) What means OMW? (line 158). What means HT? (line 211)
Answer:
All cited abbreviations (OL, HT, OMW) have been already mentioned in sections 1 and 2 of the paper.
Comment 7: Line 211-218. What means HT derivatives? LDH is a cytotoxic biomarker but not an inflammatory biomarker. Please confirm that “ex vivo” is correct.
Answer:
On L213 “and derivatives” has been removed. The expression “ex vivo” is correct as well as reported in reference [51].
Comment 8: Line 236. I cannot understand some sentences. Here, what means “the cytoprotective effects of formulations containing both syntethic HT and that derived from OMW…”? What means synthetic HT? Syntetic should be changed by synthetic.
Answer:
The sentence on L236-238 “The cytoprotective effects of formulations containing both synthetic HT and that derived from OMW was also evaluated on the same cell line but inducing toxicity by exposure for 24 hours to cadmium (Cd), mercury (Hg), and lead (Pb).” of the original paper has been rearranged as follows: “The cytoprotective effects of formulations containing both HT and OMW were compared on the same cell line by inducing toxicity after 24 hours of exposure to cadmium (Cd), mercury (Hg), and lead (Pb)”.
Comment 9: Line 366-384. This paragraph is too much extent (in a similar way to other paragraphs along the review) and should included clinical antecedents about the topic olive oil consumption and hepatic steatosis (see doi: 10.1093/jn/nxz147).
Answer:
The sentences on L368-381 “Vergani et al. in 2016 conducted an experiment in this direction [86]. In their study, the authors investigated the possible effects of polyphenols extracted from olive pomace and of the main single phenolic compounds present in the extract (tyrosol, apigenin, and oleuropein) in protecting hepatocytes against excess fat and oxidative stress. Phenolic compounds were extracted using a high pressure-temperature stirred reactor (25 bar, 180 °C for 90 min) and a mixture of ethanol/water (50:50 v/v) was used as extraction solvents yielding a total polyphenol concentration of 5.77 mg caffeic acid equivalent/mL. In order to test the biological effects of the extract, the authors exposed to a mixture of oleate/palmitate (2:1 molar ratio) for 3 h FaO cells, which represent a reliable in vitro model for hepatic steatosis. After lipid-loading, the hepatic cells were incubated for 24 h in the absence or in the presence of tyrosol, apigenin, or oleuropein (10, 13, and 50 μg/mL, respectively) after which the content of intra- and extra-cellular triglycerides (TGs) and other oxidative stress-related parameters were evaluated. The preliminary results showed that polyphenols extracted from olive pomace ameliorated lipid accumulation and lipid-dependent oxidative unbalance, showing their potential applications as therapeutic agents.” have been rewritten in “Vergani et al. carried out a study on the biological effects of polyphenols extracted from olive pomace and on the effects of single phenolic compounds present in the extract (i.e. TY, apigenin, and OL) in protecting hepatocytes against fat excess and oxidative stress [88]. The polyphenols were extracted in ethanol/water (50:50 v/v) at high pressure-temperature (25 bar, 180 °C for 90 min) obtaining a total concentration of 5.77 mg of caffeic acid equivalent/mL. In order to test the biological effects of the extract, FaO cells exposed for 3 h to a mixture of oleate/palmitate (2:1 molar ratio) were used as a model for hepatic steatosis. The cells were incubated with TY, apigenin, or OL (10, 13, and 50 μg/mL, respectively) and the content of intra- and extra-cellular triglycerides (TGs) and other oxidative stress markers measured after 24 h. The preliminary results showed that olive pomace extract ameliorated lipid accumulation and lipid-dependent oxidative unbalance, suggesting them as potential therapeutic agents.”
The sentence on L381-384 “This study is certainly of interest, but experiments on human models are essential to demonstrate that the stated effects are confirmed when the phenolic compounds are introduced as dietary supplements, establishing a clear correlation between molecules, concentration, daily dose and measurable benefit.” has been removed and substituted with “The direct correlation between a MD supplemented with EVOO and a reduced prevalence of hepatic steatosis in older individuals at high cardiovascular risk was recently investigated in a clinical trial comprising one hundred men and women (mean age: 64 ± 6 years old) at high cardiovascular risk (62% with type 2 diabetes) [89].” to support the clinical investigation reported in the section.
Comment 10: Line 409. “There is a growing interest in extracting and separating oleuropein from olive leaves because natural active compounds are safer for human health than synthetic chemicals”. I cannot understand this sentence. Why natural phenols are safer than synthetic phenols?
Answer:
We appreciated your suggestion but here in we underline how natural substances in food can be safer for human health than those of synthetic origin such as drugs, as no industrial solvents nor reagents have been used to prepare them. In this context it’s important to underline that olive leaves are not excluded from the oil production process.
Comment 11: Line 460-474. As I previously mentioned clinical studies should be included to improve the description of the healthy actions of olive oil/olive oil components. Here, the antihypertensive actions of olive components should be complemented with clinical studies (i.e. doi: 10.1007/s00394-015-1060-5).
Answer:
We thank the referee for the suggestion. The phrase “To support the key nutraceutical role of EVOO in the MD, Storniolo et al. recently analysed whether it induced changes on endothelial physiology elements, such as NO, ET-1 and ET-1 receptors, which are involved in controlling blood pressure [99].” has been written to support the clinical investigation on the antihypertensive actions of olive components.
Comment 12: Line 533. What means “human body districts”?
Answer:
The sentence on L533-534 “The anti-inflammatory effect of OLE has been studied in different human body districts and in different diseases.” of the original paper gas been rewritten as follows: “The anti-inflammatory effect of OLE has been investigated by screening all diseases in which inflammatory mechanisms are involved.”
Comment 13: Some parts of the review are too much descriptive (i.e. line 598-609). I believe that reviewed those will improve the quality of the manuscript.
Answer:
We thank the referee for the suggestion. We performed a deep revision of the English language and many parts of the manuscript (i.e. lines 598-609) were also simplified to make the article smoother and easier to read.
Comment 14: The title of the sections should be also revised. In my opinion, titles are not appropriate in same parts of the manuscript (i.e. “Kernel and seed biological activity”).
Answer:
First of all, the manuscript title has been changed as follows “Olive tree in circular economy as a source of secondary metabolites active for human and animal health beyond oxidative stress and inflammation” according to editor suggestion. In addition, section titles have been also rewritten to make them more pertinent to the contained text but also considering the whole story.
Comment 15: Along the manuscript emerge the idea that the mix compounds in by-products can be more effective than a only compound considering the possibility of an additive/synergistic effects of bioactive olive compounds as have been reported (i.e. doi: 10.1021/acs.jafc.9b04816). Perhaps this idea should be directly presented in the context of the topic.
Answer:
Experimental and clinical findings have been reported in each section of the manuscript to support the activity of by-product of olive on the human health.
Comment 16: In my opinion the review needs additional figures or revised the figures (1, 2) to improve the representation of the by-products of the olive oil industry, the quantities produced, the contamination, the different composition of each by-product, the main alternative applications of each by-product,….
Answer:
Thank the reviewer for the observation. In order to implement his recommendations and allow to read the content of the figures, figure 1 was modified by inserting data relative to the requested information:
- the percentage ratio between raw material and oil has been inserted, from which the percentage of by-products is drawn
- the icon of the olive mill has been replaced with an olive that reports the percentage mass ratio of the pulp/kernel

Reviewer 3 Report
The manuscript by Mallamaci et al ‘Olive tree in circular economy: a source of secondary metabolites active for human and animal health: not only against oxidative stress and inflammation’ is very interesting. This article has a high value. It this article has now been proven that EVOO and its by-products are an important source of bioactive compounds. The by-products, as detailed in this review, are very rich in bioactive components that currently are not yet properly exploited. The bioactive compounds can be recovered by green technologies and reused for food, agronomic, nutraceutical, and biomedical applications, in agreement with the circular economy strategy.
The article is interesting, but several issues need to be corrected and clarified.
- Line 64-104 - There is no literature source references.
- Figure 2 – There is no literature source references.
- Line 163-164 - For example, early harvested olives led to oils with a higher concentration of polyphenols, even considering the same cultivar. - Please explain this interesting issue
- Table 3 – Composition: tyrosol, caffeic acid, galactose, mannose, xylose, proteins - The numerical values should be presented with the same number of decimal places.
- Table 4 – Cultivars: Manzanilla- total polyphenols; Cultivars: Amphissis, Amigdalolia and Leccino: total polyphenols and DPPH; Cultivars: Leccino - FRAP - The numerical values should be presented with the same number of decimal places. They should be accurate to 2 decimal places.
- Line 307 - Nunes et al. investigated the use of olive pomace … - There is no literature source references.
- Briante, R.; La Cara, F.; Febbraio, F.; Patumi, M.; Nucci, R. Bioactive derivatives from oleuropein by a biotransformation on Olea europaea leaf extracts. J. Biotechnol. 2002, 93, 109–119 – No citation in the text.
Author Response
Answer to Comments and Suggestions for Authors
Reviewer 3
Comment: The manuscript by Mallamaci et al ‘Olive tree in circular economy: a source of secondary metabolites active for human and animal health: not only against oxidative stress and inflammation’ is very interesting. This article has a high value. It this article has now been proven that EVOO and its by-products are an important source of bioactive compounds. The by-products, as detailed in this review, are very rich in bioactive components that currently are not yet properly exploited. The bioactive compounds can be recovered by green technologies and reused for food, agronomic, nutraceutical, and biomedical applications, in agreement with the circular economy strategy.
The article is interesting, but several issues need to be corrected and clarified.
Answer:
We thank the reviewer for the comment. The whole manuscript has been revised. Several sentences have been rewritten and/or clarified. The bibliography has been integrated with more recent literature, while references that were poorly congruent with the main topic have been eliminated. The style has been revised and the plagiarized sentences eliminated.
Comment 1: Line 64-104 - There is no literature source references.
Answer:
About lines 64-104, the sentences on L64-80 “Circular economy has gained increasing importance as a tool capable of supporting the challenges of sustainable development to accelerate the implementation of the “Transforming our world: the 2030 Agenda for Sustainable Development”. The concept of circular economy responds to the desire for sustainable growth, in the context of the increasing pressure to which production and consumption are subjecting world resources and the environment. The circular economy is based on a fundamental paradigm shift. The economic and ecological systems are no longer at the same level as they were in traditional economic analysis, where natural resources, factors of production, economic goods and services, waste and scrap were exchanged. Unlike in the linear system, which starts with raw materials and ends with waste, the circular economy is an economy in which today’s products are tomorrow’s resources, in which the value of materials is maintained or recovered as much as possible, minimizing waste and environmental impact.
The transition to circular economy requires a cultural and structural change: a deep revision and innovation of production, distribution, and consumption models. These are the cornerstones of this change, with the abandonment of the linear economy, the overcoming of the recycling economy, and the arrival of the circular economy, passing through new business models and transformation of waste into resources with high added value.” have been rewritten as follows: “However, in coherence with the “circular economy” principle, it is important to valorise these waste products, containing high levels of secondary metabolites, thus accelerating the implementation of the “Transforming our world: the 2030 Agenda for Sustainable Development” [10,11].
Nevertheless, the transition from linear to circular economy requires a cultural and structural change: a deep revision and innovation of production, distribution, and consumption models [12].
Literature source references have been reported to support the text content.
Comment 2: Figure 2 – There is no literature source references. LISA
Answer:
We thank the reviewer for the observation. In order to implement his recommendations and make easier to read their content, figure 1 was also modified by inserting data relative to the requested information:
- the percentage ratio between raw material and oil has been inserted, from which the percentage of by-products is drawn
- the icon of the olive mill has been replaced with an olive that reports the percentage mass ratio of the pulp/kernel
Comment 3: Line 163-164 - For example, early harvested olives led to oils with a higher concentration of polyphenols, even considering the same cultivar. - Please explain this interesting issue
Answer:
This issue is strictly correlated with the sentence on L159-162 of the original paper “Many production factors (e.g. cultivar, ripening time, and extraction method) as well as environmental factors (e.g. climate, precipitations, and age of the trees) are responsible of different content and composition of polyphenols in oil [22].”. To avoid confusion, the statement has been now removed leaving to the reader the opportunity to deepen the discussion following the cited reference [22].
Comment 4: Table 3 – Composition: tyrosol, caffeic acid, galactose, mannose, xylose, proteins - The numerical values should be presented with the same number of decimal places.
Answer:
Tables numbering have been changed because Table 1 has been removed in according to editor suggestion. The numerical values on the table have been presented with the same number of decimal places.
Comment 5: Table 4 – Cultivars: Manzanilla- total polyphenols; Cultivars: Amphissis, Amigdalolia and Leccino: total polyphenols and DPPH; Cultivars: Leccino - FRAP - The numerical values should be presented with the same number of decimal places. They should be accurate to 2 decimal places.
Answer:
The numerical values on Table 3 have been now corrected with the same number of decimal places.
Comment 6: Line 307 - Nunes et al. investigated the use of olive pomace … - There is no literature source references.
Answer:
We thank the referee for the suggestion, the literature source (ref. [83]) was at the end of the paragraph and it has now moved to the first sentence.
Comment 7: Briante, R.; La Cara, F.; Febbraio, F.; Patumi, M.; Nucci, R. Bioactive derivatives from oleuropein by a biotransformation on Olea europaea leaf extracts. J. Biotechnol. 2002, 93, 109–119 – No citation in the text.
Answer:
We thank the referee for spotting the error. The reference Briante, R.; La Cara, F.; Febbraio, F.; Patumi, M.; Nucci, R. Bioactive derivatives from oleuropein by a biotransformation on Olea europaea leaf extracts. J. Biotechnol. 2002, 93, 109–119 has been removed because it is not necessary in this version of the manuscript.

Reviewer 4 Report
The article entitled "Olive tree in circular economy: a source of secondary metabolites active for human and animal health: not only against oxidative stress and inflammation" is very interesting in the field of "food by products". The circular economy is gaining momentum due to the great benefits it brings to society. The work covers many aspects of olive tree residues, the bibliography is updated. However, there are aspects that should be improved for the article to be accepted.
- First of all the meaning of abbreviations is sometimes not found.
- The terms "in vivo" and "in vitro" must be in italics.
- In the composition of olive oil the compound is not mentioned is squalene, which has very interesting therapeutic properties.
- Many sections are not clear, they should have a summary table with the applications, the dose used and the reference. Similar to table 6.
Author Response
Answer to Comments and Suggestions for Authors
Reviewer 4
Comment: The article entitled "Olive tree in circular economy: a source of secondary metabolites active for human and animal health: not only against oxidative stress and inflammation" is very interesting in the field of "food by products". The circular economy is gaining momentum due to the great benefits it brings to society. The work covers many aspects of olive tree residues, the bibliography is updated. However, there are aspects that should be improved for the article to be accepted.
- First of all the meaning of abbreviations is sometimes not found.- The terms "in vivo" and "in vitro" must be in italics.- In the composition of olive oil the compound is not mentioned is squalene, which has very interesting therapeutic properties.
- Many sections are not clear, they should have a summary table with the applications, the dose used and the reference. Similar to table 6.
Answer:
We thank the reviewer for the comment. We focused our attention on each section of the paper with the aim to simplify and to clarify the content of the text. The bibliography has been integrated with more recent literature. The style and the English language have been deeply revised and the plagiarized sentences eliminated.
Comment 1: First of all the meaning of abbreviations is sometimes not found.
Answer:
We checked the entire document abbreviations, the full names have been now reported in the manuscript.
Comment 2: The terms "in vivo" and "in vitro" must be in italics.
Answer:
As suggested, the terms "in vivo" and "in vitro" have been rewritten in italics.
Comment 3: In the composition of olive oil the compound is not mentioned is squalene, which has very interesting therapeutic properties.
Answer:
The composition of squalene (as well as other minor compounds) in the olive by-products is not mentioned due its variables quantities. Furthermore, Table 1 has been removed from the original paper in according to editor suggestion: “there is no need for such a detailed table, all compounds present in the table cannot be recycled or are found in variables quantities.”
Comment 4: Many sections are not clear, they should have a summary table with the applications, the dose used and the reference. Similar to table 6.
Answer:
The manuscript has been carefully revised in all its section.

Round 2
Reviewer 1 Report
The authors should check this manuscript more carefully, and revise all of the errors to improve this paper.
Line 33, “ethyl acetate extraction” should be revised as “ethyl acetate extract”
- Title of Table 1, “[13-21]” should be revised as “[13-21]”.
- Page 5, reference 17 was disappeared.
- Ref. 18, 22, 25, 27, 28, 31, 42, 45, 51, 53, 54, 68, 85, 95, 103, and 123 should be followed the journal format and uniform.
Author Response
Response to Reviewer 1 Comments
Comment 1: The authors should check this manuscript more carefully, and revise all of the errors to improve this paper.
Response 1:
The whole manuscript has been carefully read and all revisions have been clearly highlighted by using the "Track Changes" function in Microsoft Word document, to make them easily visible to the reviewers as suggested by Editor.
Comment 2: Line 33, “ethyl acetate extraction” should be revised as “ethyl acetate extract” Line 33
Response 2:
The sentence “ethyl acetate extraction” is never reported for the whole manuscript
Comment 3: Title of Table 1, “[13-21]” should be revised as “[13-21]”.
Response3:
The references in the table 1 have been corrected as suggested by the Reviewer
Comment 4: Page 5, reference 17 was disappeared.
Response 4:
Thanks to the Review for this observation. We have replaced the reference 15 with the one 17. The first has been erroneously reported in the line 119 of the manuscript.
Comment 5: Ref. 18, 22, 25, 27, 28, 31, 42, 45, 51, 53, 54, 68, 85, 95, 103, and 123 should be followed the journal format and uniform.
Response 5:
Thanks to the Reviewer for this observation. All the mentioned references have been rewritten by following the journal format.

Reviewer 2 Report
The revised version of the manuscript entitled “Olive tree in circular economy as a source of secondary metabolites active for human and animal health beyond oxidative stress and inflammation” submitted to Molecules by Dr. Mallamaci and co-workers has considered my comments/suggestions but a few considerations/changes should be included before to be finish this review phase.
Fat intake is a risk factor to colorectal cancer (cited in line 142) and oleic acid appears as proliferative fatty acid on colorectal cancer cell growth (see doi: 10.3389/fphar.2020.529976) but recently Storniolo and co-workers demonstrated that is effect of oleic acid is reverted in the presence of different olive oil minor compounds (doi: 10.1021/acs.jafc.9b04816) and these authors reported that olive oil added to some dishes such as sofrito regulates colorectal cancer cell line growth (doi: 10.1021/acsomega.9b04329). I believe that these relevant information’s could be included in the manuscript (line 142?). These aspects support that extra virgin olive oil can have healthy effects whereas high oleic sunflower oil does not present these actions.
Author Response
Response to Reviewer 2 Comments
Comment 1: The revised version of the manuscript entitled “Olive tree in circular economy as a source of secondary metabolites active for human and animal health beyond oxidative stress and inflammation” submitted to Molecules by Dr. Mallamaci and co-workers has considered my comments/suggestions but a few considerations/changes should be included before to be finish this review phase.
Fat intake is a risk factor to colorectal cancer (cited in line 142) and oleic acid appears as proliferative fatty acid on colorectal cancer cell growth (see doi: 10.3389/fphar.2020.529976) but recently Storniolo and co-workers demonstrated that is effect of oleic acid is reverted in the presence of different olive oil minor compounds (doi: 10.1021/acs.jafc.9b04816) and these authors reported that olive oil added to some dishes such as sofrito regulates colorectal cancer cell line growth (doi: 10.1021/acsomega.9b04329). I believe that these relevant information’s could be included in the manuscript (line 142?). These aspects support that extra virgin olive oil can have healthy effects whereas high oleic sunflower oil does not present these actions.
Response 2
We thank the Reviewer for this important observation.
The paragraph on Lines146-150 “Recently, Storniolo and co-workers demonstrated that the role of oleic acid in the colon cancer cells growth is reverted in the presence of olive oil representative minor components, suggesting that the consumption of seed oils, high oleic acid seed oils, or olive oil will probably have different effects on colorectal cancer (doi: 10.1021/acs.jafc.9b04816, doi: 10.1021/acsomega.9b04329).”has been written to support the benefits of olive oil in human health. References [35,36] have been reported.

Reviewer 4 Report
I think the authors have improved the work. Only minor revision so that the article can be accepted. In table 5, I believe that the biological effects of olive leaves extracts should show the corresponding references.
Author Response
Response to Reviewer 4 Comments
Comment 1: In table 5, I believe that the biological effects of olive leaves extracts should show the corresponding references.
Response 2
We thank the Reviewer for this suggestion. Table 5 has been removed in the previous revision. We suppose the Reviewer refers to the Table 4. Then the corresponding references of the biological effects of olive leaves extracts have been introduced in Table 4.

This manuscript is a resubmission of an earlier submission. The following is a list of the peer review reports and author responses from that submission.
Round 1
Reviewer 1 Report
The authors give a nice comprehension about compounds usually found in olives, mainly focussing on the olive trees of their own home country. Many compounds are listed, but their biological/medicinal effects are presented in a very general way, and very often details are missing. There would be a fast amount of data available (there are books on that!) but the review remains superficial and at some paragraphs it reads like an article in a newspaper or a magazine. I'd like to recommend not to publish this manuscript.
Reviewer 2 Report
The article is a nice piece of work. Olive oil mills wastewaters are also used after some treatment as feed for productive animals like sheep, pork and chicken. Please create a small chapter around one page with some references. Some of them are shown here:
- Kerasioti et al. Toxicology Rep[orts , 2017
- Makri et al. In Vivo, 2018
- Gerasopoulos K. et al. Food and Chemical Toxicology, 2015
Reviewer 3 Report
This literature review encompasses a broad paper with information from olive tree (Olea europaea) and its by-products on secondary metabolites bioactivities. Under this present form, the paper appears to have numerous weaknesses such the title didn’t reflect this review article and many sentences were verbatim plagiarism. Nonetheless, the authors should revise the followings before submitting the article:
- Many review articles of this topic have been published. The relative similar review articles were published in the J Sci Food Agric. and (Olivetree (Olea europaea L.) leaf as a waste by-product of table olive and olive oil industry: a review. J Sci Food Agric. 2018, 98:1271-1279; Effects of olive oil and its minor components on cardiovascular diseases, inflammation, and gut microbiota. Nutrients. 2019, 11:1826)
- The authors sited information from various papers but did not organize, comment, or contrast the findings.
- In this text, many sentences were verbatim plagiarism from references (Molecules . 2017, 22:1858; Crit Rev Food Sci Nutr., 2018, 58, 2829-2841; Antioxidants, 2018, 7, 170; Int J Mol Sci. 2018,19, 2305……etc.).
- There were many errors throughout the paper therefore the paper should be carefully checked. a. Ex:Abstract section: Please provide the full name of EVOO and EFSA; b. Kywords section: : Olea Europea---à Olea europaea L.; Olive mill waste water (OMV)??; c. Content of subtitle should be followed the journal format (upper case) and uniform. (ex: 6. Biological activities of Olive oil secondary metabolites; 9. Olive leaves Extract: production and composition; 10.1 Cardioprotective Effect; 10.2 Neurodegenerative diseases…….etc.); d. Line 72, “composition [1,2].Both the constituents” ……..> “composition [1,2]. Both the constituents”; line 170, “systems [11].Phenolic” …..>“systems [11]. Phenolic” ; line 178, “Hydroxytyrosol (HT)”……>“hydroxytyrosol (HT)”; line 187, “O. Capensis, O. Dioica, O. Brachiata and O. Obovata” ……>“O. capensis, O. dioica, O. brachiata and O. obovata”…….etc.